# Natural variation in stochastic photoreceptor specification and color preference in *Drosophila*

**Caitlin Anderson[1], India Reiss[1], Cyrus Zhou[1†], Annie Cho[1†], Haziq Siddiqi[1†], Benjamin Mormann[2], Cameron M Avelis[3], Peter Deford[1], Alan Bergland[4], Elijah Roberts[3], James Taylor[1], Daniel Vasiliauskas[5], Robert J Johnston[1]\***

[1]Department of Biology, Johns Hopkins University, Baltimore, United States; [2]Center for Developmental Genetics, Department of Biology, New York University, New York, United States; [3]Department of Biophysics, Johns Hopkins University, Baltimore, United States; [4]Department of Biology, University of Virginia, Charlottesville, United States; [5]Paris-Saclay Institute of Neuroscience, Université Paris Sud, Centre National de la Recherche Scientifque, Université Paris-Saclay, Gif-sur-Yvette, France

**\*For correspondence:**
robertjohnston@jhu.edu

[†]These authors contributed equally to this work

**Competing interests:** The authors declare that no competing interests exist.

**Abstract** Each individual perceives the world in a unique way, but little is known about the genetic basis of variation in sensory perception. In the fly eye, the random mosaic of color-detecting R7 photoreceptor subtypes is determined by stochastic on/off expression of the transcription factor Spineless (Ss). In a genome-wide association study, we identified a naturally occurring insertion in a regulatory DNA element in *ss* that lowers the ratio of $Ss^{ON}$ to $Ss^{OFF}$ cells. This change in photoreceptor fates shifts the innate color preference of flies from green to blue. The genetic variant increases the binding affinity for Klumpfuss (Klu), a zinc finger transcriptional repressor that regulates *ss* expression. Klu is expressed at intermediate levels to determine the normal ratio of $Ss^{ON}$ to $Ss^{OFF}$ cells. Thus, binding site affinity and transcription factor levels are finely tuned to regulate stochastic expression, setting the ratio of alternative fates and ultimately determining color preference.
DOI: https://doi.org/10.7554/eLife.29593.001

## Introduction

Organisms require a diverse repertoire of sensory receptor neurons to perceive a range of stimuli in their environments. Differentiation of sensory neurons often requires stochastic mechanisms whereby individual neurons randomly choose between different fates. Stochastic fate specification diversifies sensory neuron subtypes in a wide array of species including worms, flies, mice, and humans (*Ressler et al., 1993*; *Roorda and Williams, 1999*; *Troemel et al., 1999*; *Hofer et al., 2005*; *Johnston and Desplan, 2010*; *Magklara and Lomvardas, 2013*; *Alqadah et al., 2016*; *Viets et al., 2016*). How naturally occurring changes in the genome affect stochastic mechanisms to alter sensory system development and perception is poorly understood. To address this question, we investigated natural variation in stochastic color photoreceptor specification in the *Drosophila* retina.

The fly eye, like the human eye, contains a random mosaic of photoreceptors defined by expression of light-detecting Rhodopsin proteins (*Montell et al., 1987*; *Bell et al., 2007*; *Johnston and Desplan, 2010*; *Viets et al., 2016*). In flies, the stochastic on/off expression of Spineless (Ss), a PAS-bHLH transcription factor, determines R7 photoreceptor subtypes. Ss expression in a random subset of R7s induces 'yellow' (yR7) fate and expression of Rhodopsin4 (Rh4), whereas the absence of Ss in the complementary subset of R7s allows for 'pale' (pR7) fate and Rhodopsin3 (Rh3) expression

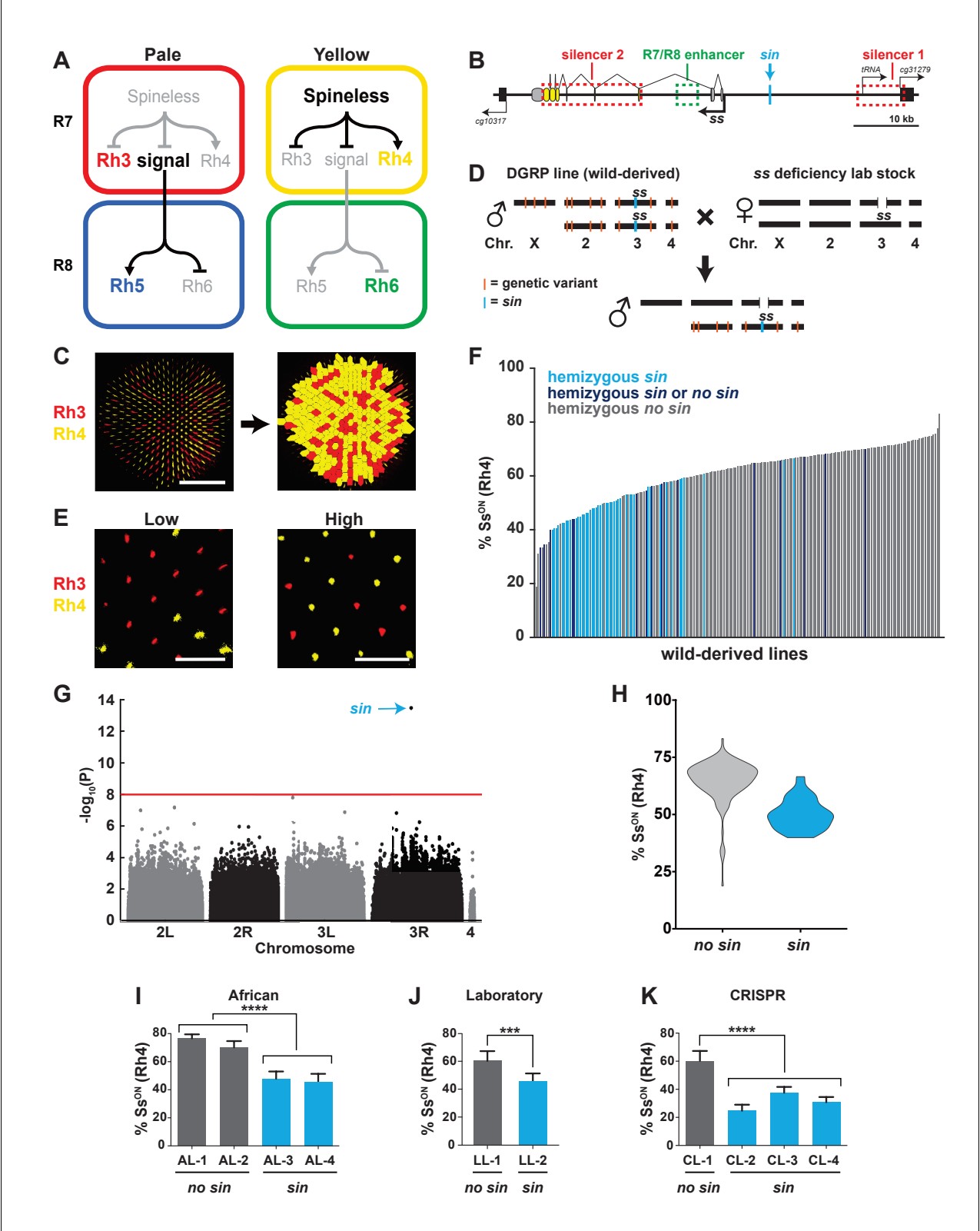

**Figure 1.** A naturally-occurring single base insertion (*sin*) in the *ss* locus lowered the ratio of Ss[ON] to Ss[OFF] R7s. (**A**) R7 and R8 subtypes are determined by the on/off expression of Spineless (Ss). (Left) The absence of Ss allows Rh3 expression in pale R7s and Rh5 expression in pale R8s. (Right) Expression of Ss induces Rh4 expression in yellow R7s and Rh6 expression in yellow R8s. The signal by which Spineless mediates Rh5 vs. Rh6 expression in R8s is currently unknown. (**B**) Schematic of the *ss* locus. Green dashed rectangle indicates *R7/R8 enhancer*; red dashed rectangles indicate *silencer 1* and

*Figure 1 continued on next page*

*Figure 1 continued*

*silencer 2*; blue line indicates Klu binding site; blue arrow indicates *ss insertion/sin;* gray ovals represent untranslated exons; yellow ovals represent translated exons; black boxes indicate neighboring genes; arrows indicate transcriptional starts. See also *Figure 1—figure supplement 1*. (C) Image of a whole mount fly retina. (Left) Stochastic distribution of R7s expressing Rh3 (Ss$^{OFF}$) or Rh4 (Ss$^{ON}$). Scale bar indicates 100 µm. (Right) An automated counting system identified and counted Rh3- and Rh4-expressing R7s. (D) Crossing scheme: Wild-derived DGRP flies were crossed with *ss* deficiency flies, yielding progeny that were hemizygous at the *ss* locus. Orange lines indicate hypothetical genetic variants; blue line indicates *sin* in *ss*. (E) Representative images from progeny in (D) with low (left; DGRP-397) and high (right; DGRP-229) proportions of Ss$^{ON}$ (Rh4) R7s. Scale bar indicates 20 µm. (F) Ss$^{ON}$ proportion varied across DGRP fly lines. *sin* was enriched in lines with a low proportion of Ss$^{ON}$ R7s. Each bar represents progeny from a single DGRP line, and bars are arranged in rank order. Light blue bars indicate hemizygous *sin*. Dark blue bars indicate hemizygous *sin* or hemizygous *no sin* (original DGRP line was heterozygous *sin/no sin*). Gray bars indicate hemizygous *no sin*. See also *Figure 1—source data 1*. (G) GWAS identified *sin* as a genetic variant associated with Ss expression. Manhattan plot of the genetic variant p-values. Genetic variants above the red line (Bonferroni correction) are considered significant. Arrow indicates *sin*. (H) *sin* was enriched in lines with a low proportion of Ss$^{ON}$ R7s. Violin plot of DGRP lines with and without *sin*. (I–K) Flies with *sin* displayed a lower proportion of Ss$^{ON}$ R7s compared to flies without *sin*. AL indicates African lines; LL indicates laboratory lines; CL indicates lines in which *sin* was inserted with CRISPR. **** indicates p<0.0001; *** indicates p<0.001. Error bars indicate standard deviation (SD). See also *Figure 1—figure supplements 2–4*.

DOI: https://doi.org/10.7554/eLife.29593.002

The following source data and figure supplements are available for figure 1:

**Source data 1.** DGRP % Ss$^{ON}$ phenotypes.
DOI: https://doi.org/10.7554/eLife.29593.007
**Figure supplement 1.** The regions encompassing and neighboring the Klu binding site have transcriptional activity in the eye.
DOI: https://doi.org/10.7554/eLife.29593.003
**Figure supplement 2.** *sin* and *klu* genetic perturbations alter the proportion of Ss$^{ON}$ R7s.
DOI: https://doi.org/10.7554/eLife.29593.004
**Figure supplement 3.** Analysis of *sin* allele frequency.
DOI: https://doi.org/10.7554/eLife.29593.005
**Figure supplement 4.** *klu* and *sin* genetic perturbations do not affect levels of Ss expression in Ss$^{ON}$ R7s.
DOI: https://doi.org/10.7554/eLife.29593.006

(*Figure 1A*) (*Wernet et al., 2006*; *Johnston et al., 2011*; *Thanawala et al., 2013*; *Johnston and Desplan, 2014*). The on/off state of Ss in a given R7 also indirectly determines the subtype fate of the neighboring R8 photoreceptor. **p**R7s lacking Ss signal to **p**R8s to activate expression of blue-detecting Rhodopsin5 (Rh5). **y**R7s expressing Ss do not send this signal, resulting in expression of green-detecting Rhodopsin6 (Rh6) in **y**R8s (*Figure 1A*) (*Franceschini et al., 1981*; *Montell et al., 1987*; *Zuker et al., 1987*; *Chou et al., 1996*; *Huber et al., 1997*; *Chou et al., 1999a*; *Mikeladze-Dvali et al., 2005*; *Mazzoni et al., 2008*; *Vasiliauskas et al., 2009*; *Jukam and Desplan, 2011*; *Hsiao et al., 2013*; *Johnston, 2013*; *Jukam et al., 2013*; *Jukam et al., 2016*; *Yan et al., 2017*).

The stochastic decision to express Ss is made cell-autonomously at the level of the *ss* gene locus via a random repression mechanism. The *R7/R8 enhancer* induces *ss* expression in all R7s, whereas two silencer regions (*silencer 1* and *2*) repress expression in a random subset of R7s (*Figure 1B*) (*Johnston and Desplan, 2014*).

Though the stochastic expression of Ss is binary (i.e. on or off) in individual R7s, it does not result in a simple 50:50 on/off ratio across the population of R7s in a given retina. In most lab stocks, Ss is on in ~65% of R7s and off in ~35% (*Figure 1C*) (*Wernet et al., 2006*; *Johnston and Desplan, 2014*). Here, we find that the proportion of Ss$^{ON}$ to Ss$^{OFF}$ R7s varies greatly among fly lines derived from the wild. We performed a genome-wide association study (GWAS) and identified a single base pair insertion that increases the affinity of a DNA binding site for a transcriptional repressor, significantly reducing the Ss$^{ON}$/Ss$^{OFF}$ ratio. This genetic variant changes the proportion of photoreceptor sub-types and alters the innate color preference of flies.

## Results

### *sin* decreases the ratio of Ss$^{ON}$ to Ss$^{OFF}$ R7s

To determine the mechanism controlling the ratio of stochastic on/off Ss expression, we analyzed the variation in 203 naturally-derived lines collected from Raleigh, North Carolina (Drosophila Genetic Reference Panel (DGRP)) (*Mackay et al., 2012*). We evaluated Rh4 and Rh3 expression, as

they faithfully report Ss expression in R7s (i.e. $Ss^{ON}$ = Rh4; $Ss^{OFF}$ = Rh3) (*Figure 1A*) (*Thanawala et al., 2013*; *Johnston and Desplan, 2014*). To facilitate scoring, we generated a semi-automated counting system to determine the Rh4:Rh3 ratio for each genotype (*Figure 1C*).

To assess the variation in the DGRP lines attributable to the *ss* locus and limit the phenotypic contribution of recessive variants at other loci, we crossed each DGRP line to a line containing a ~200 kb deficiency covering the *ss* locus and analyzed Rh3 and Rh4 expression in the F1 male progeny (*Figure 1D*). This genetic strategy generated flies hemizygous (i.e. single copy) for the wild-derived *ss* gene locus, heterozygous wild-derived/lab stock for the second, third, and fourth chromosomes, and hemizygous lab stock for the X chromosome (*Figure 1D*). While the lab stock expressed Ss (Rh4) in 62% of R7s under these conditions, expression among the DGRP lines varied significantly, ranging from 19% to 83% $Ss^{ON}$ (Rh4) (*Figure 1E–F*; *Figure 1—source data 1*).

To identify the genetic basis of this variation, we performed a genome-wide association study (GWAS) using the $Ss^{ON}$ (Rh4) phenotype data and inferred full genome sequences of the progeny of each DGRP line crossed with the *ss* deficiency line. We performed an association analysis and identified a single base pair insertion within the *ss* locus ('*ss* insertion' or '*sin*') that was significant ($p < 10^{-13}$) after Bonferroni correction (*Figure 1G*). *sin* was enriched in DGRP lines with a low ratio of $Ss^{ON}$ to $Ss^{OFF}$ R7s (*Figure 1F and H*).

We next confirmed the regulatory role of *sin*. Naturally derived lines from Africa that are homozygous for *sin* displayed a decrease in the proportion of $Ss^{ON}$ (Rh4) R7s compared to lines from Africa lacking *sin* (*Figure 1I*) (*Lack et al., 2015*). We identified *sin* on a balancer chromosome (*TM6B*) in a lab stock that similarly displayed a decrease in the proportion of $Ss^{ON}$ (Rh4) R7s when *ss* was hemizygous (*Figure 1J*). To definitively test the role of *sin*, we used CRISPR to insert *sin* into a lab stock. Flies hemizygous for CRISPR *sin* alleles displayed a significant decrease in the proportion of $Ss^{ON}$ (Rh4) R7s (*Figure 1K*). Using a Ss antibody, we examined Ss expression directly and found that flies homozygous for CRISPR *sin* alleles displayed a significant decrease in the proportion of $Ss^{ON}$ R7s (*Figure 1—figure supplement 2A–B and E*). Thus, *sin* causes a decrease in the ratio of $Ss^{ON}$ to $Ss^{OFF}$ R7s.

## *sin* shifts innate color preference from green to blue

As *sin* alters the proportion of color-detecting photoreceptors, we hypothesized that it would also change color detection and preference. When presented with two light stimuli in a T-maze (*Tully and Quinn, 1985*), flies will phototax toward the light source that they perceive as more intense (*Figure 2A*) (*McEwen, 1918*; *Heisenberg and Wolf, 1984*; *Choe and Clandinin, 2005*). The absorption spectra of Rh3 and Rh4 significantly overlap in the UV range (*Feiler et al., 1992*), complicating behavioral assessment of color preference caused by differences in R7 photoreceptor ratios. Instead, we focused on the perception of blue light by Rh5 and green light by Rh6 in the R8 photoreceptors, as these Rhodopsins have more distinct absorption spectra (*Salcedo et al., 1999*). Because R8 fate is coupled to R7 fate (*Chou et al., 1996*) (*Figure 1A*), we predicted that flies with *sin* would have a low ratio of Rh6- to Rh5-expressing R8s and would consequently prefer blue light, while flies without *sin* would have a higher ratio of Rh6- to Rh5-expressing R8s and would instead prefer green light. Indeed, DGRP lines containing *sin* preferred blue light, while DGRP lines lacking *sin* preferred green light (*Figure 2A–C*; *Figure 2—source data 1*), showing that *sin* changes innate color preference in flies. Natural variation in the gene regulatory pathway controlling Rh5 and Rh6 expression downstream of Ss (*Mikeladze-Dvali et al., 2005*; *Jukam and Desplan, 2011*; *Jukam et al., 2013*; *Viets et al., 2016*) or in the neural circuit downstream of R8 signaling likely caused the green light preference of some lines with *sin*.

## *sin* increases the binding affinity for the Klumpfuss transcription factor

*sin* is a single base pair insertion within a previously uncharacterized non-coding region of the *ss* locus located ~7 kb upstream of the transcriptional start (*Figure 1B* and *Figure 3A*). To identify *trans* factors whose binding might be affected by *sin*, we searched for binding motifs affected by *sin* in SELEX-seq (*Nitta et al., 2015*) and bacterial one-hybrid datasets (B1H) (*Zhu et al., 2011*; *Enuameh et al., 2013*). *sin* lies in a predicted binding site for the zinc finger transcription factor Klumpfuss (Klu), the fly homolog of Wilms' Tumor Suppressor Protein 1 (WT1) (*Figure 3B*, *Figure 3—*

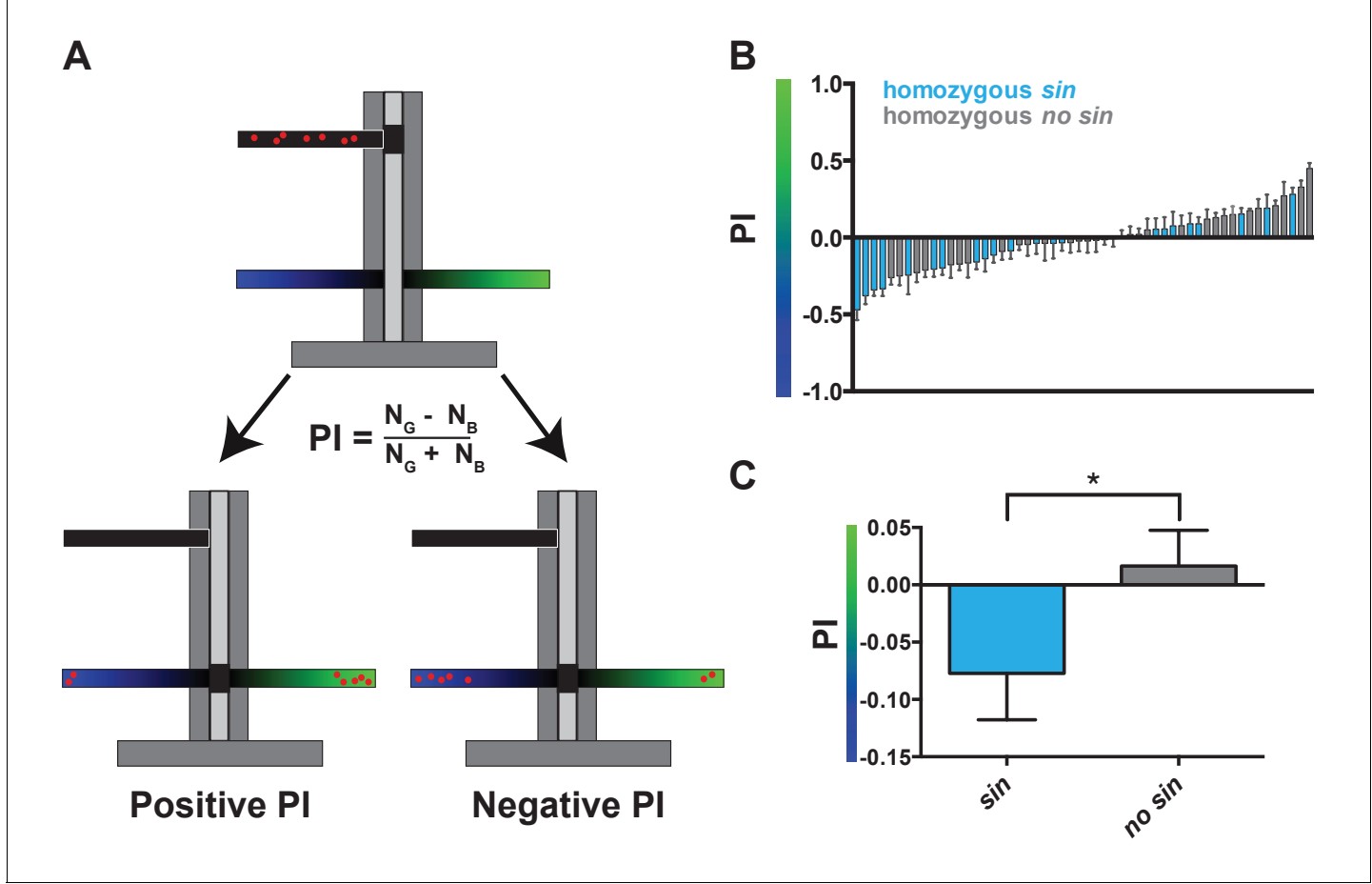

**Figure 2.** Wild-derived flies with *sin* display a shift in innate color preference from green to blue. (**A**) Schematic of T-maze apparatus. Red dots represent flies. PI = Preference Index. Positive PI indicates preference for green light; negative PI indicates preference for blue light. $N_G$: number of flies on green side; $N_B$: number of flies on blue side. (**B–C**) Flies with *sin* preferred blue light, while flies without *sin* preferred green light. Color of bar indicates genotype of DGRP line. Light blue bars indicate homozygous *sin*. Gray bars indicate homozygous *no sin*. Error bars indicate standard error of the mean (SEM). See also *Figure 2—source data 1*. (**B**) PIs for individual DGRP lines with and without *sin*. (**C**) Averages of PIs for DGRP lines with (n = 23) and without *sin* (n = 31). * indicates p<0.05.
DOI: https://doi.org/10.7554/eLife.29593.008

The following source data is available for figure 2:

**Source data 1.** DGRP behavior phenotypes.
DOI: https://doi.org/10.7554/eLife.29593.009

figure supplement 1A) (*Klein and Campos-Ortega, 1997*; *Yang et al., 1997*). This region is perfectly conserved across 21 *Drosophila* species covering 50 million years of evolution, consistent with a critical regulatory role (*Figure 3C*, *Figure 3—figure supplement 1B–C*).

To evaluate the effect of *sin* on Klu binding, we analyzed available SELEX-seq binding data (*Nitta et al., 2015*), focusing on the core 10-mer. The number of reads containing the Klu binding site with *sin* (CGCCCACAC**C**) was significantly higher than without *sin* (CGCCCACAC**A**) (*Figure 3D*), and thus, Klu binds the endogenous *ss* sequence with *sin* better than without it. Considering the frequency of 10-mers as a measure of site preference, we found that 506 10-mers (0.10%) have frequencies greater than the Klu site without *sin,* whereas only 366 10-mers (0.07%) have frequencies greater than the Klu site with *sin*. Together, *sin* increases the binding affinity of the Klu site in vitro.

We further analyzed SELEX-seq data to understand the differences between Klu binding affinities for the predicted optimal site and endogenous site in *ss*. The endogenous 10-mer core sequence in *ss* (CGCCCAC**A**CA) deviates from the optimal site (CGCCCAC**G**CA) at position 8, causing a dramatic reduction in the affinity for the endogenous site (*Figure 3E*). Interestingly, *sin* in position 10

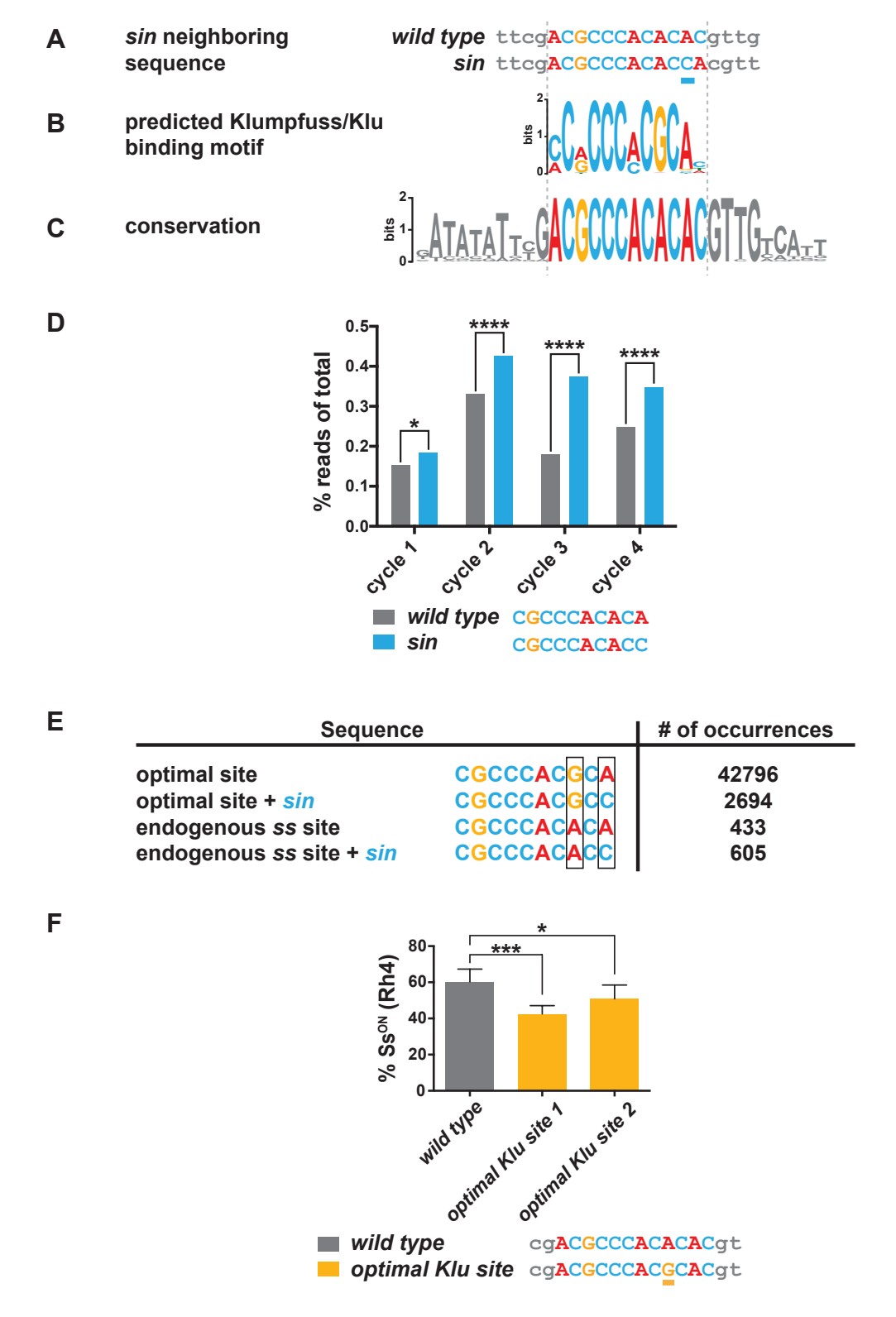

**Figure 3.** *sin* increases the binding affinity for the transcription factor Klumpfuss. (A–C) Colored bases indicate the predicted Klu binding site. (A) *sin* is a single base pair insertion of a C at Chr. 3R: 16,410,775 (release 6). Underline indicates *sin*. (B) Position weight matrix (PWM) for Klu binding predicted from SELEX-seq. See also *Figure 3—figure supplement 1A*. (C) The Klu site is perfectly conserved across 21 species of *Drosophila* covering 50 million years of evolution. Conservation logo of the Klu site and neighboring sequence in the *ss* locus. Height of bases indicates degree of conservation. See
*Figure 3 continued on next page*

*Figure 3 continued*

also *Figure 3—figure supplement 1B*. (D) *sin* increased Klu binding affinity in vitro. Quantification of the number of reads for the Klu site with and without *sin* in four cycles of SELEX-seq. * indicates p<0.05; **** indicates p<0.0001. (E) Comparing the number of read-occurrences in the SELEX-seq dataset, *sin* increases binding of Klu to the endogenous *ss* site but decreases binding of Klu to the optimal site, suggesting a binding site affinity dependence between bases. Boxes highlight positions 8 and 10 in the 10-mer core sequences. (F) Flies hemizygous for an optimal Klu site displayed a lower proportion of Ss$^{ON}$ R7s compared to wild type flies. *optimal Klu site 1* and *optimal Klu site 2* are independent lines derived from CRISPR-mediated mutagenesis. * indicates p<0.05; *** indicates p<0.001. Error bars indicate standard deviation (SD).

DOI: https://doi.org/10.7554/eLife.29593.010

The following figure supplement is available for figure 3:

**Figure supplement 1.** Consensus Klu binding sites and conservation of the Klu site in the *ss* locus.

DOI: https://doi.org/10.7554/eLife.29593.011

significantly decreases binding affinity for the optimal site (compare CGCCCACGC<u>A</u> to CGCCCACGC<u>C</u>), whereas it increases affinity for the endogenous Klu site in *ss* (compare CGCCCA-CAC<u>A</u> to CGCCCACAC<u>C</u>) (*Figure 3E*). A PWM is a good representation of the sequence preferences of a DNA-binding protein, but it assumes independent contributions of individual bases. In this case, we observe dependence between positions within the motif that the PWM disregards. Our analysis indicates that Klu binding affinity is dependent on the relationship between the bases in position 8 and 10. This dependence reveals that Klu preferentially interacts with the site with *sin* (C in position 10) over the site without *sin* (A in position 10) in the endogenous *spineless* locus (A in position 8), in contrast to the general predictions of the PWM (preferred G in position eight and A in position 10). Dependence between positions suggests that binding of transcription factors like Klu is determined not only by sequence but also by DNA shape, as has been described previously (*Abe et al., 2015*; *Zhou et al., 2015*; *Chiu et al., 2017*). These data suggest that the Klu site in the endogenous locus is a low-affinity site and that *sin* increases its affinity.

Since *sin* is predicted to increase the binding affinity for Klu and *sin* caused a reduction in the on/off ratio of Ss expression, we hypothesized that mutating the Klu site to an optimized high-affinity site would also cause a decrease in the proportion of Ss$^{ON}$ R7s. We used CRISPR to mutate the endogenous low-affinity Klu site (ACGCCCAC<u>**A**</u>CAC) to the predicted optimized high-affinity site (ACGCCCAC<u>**G**</u>CAC) and observed a decrease in the proportion of Ss$^{ON}$ R7s similar to flies with *sin* (*Figure 3F*). The observation that an optimized high-affinity Klu site causes a similar phenotype as *sin* is consistent with the conclusion that *sin* increases the binding affinity for Klu.

## Klu lowers the Ss$^{ON}$/Ss$^{OFF}$ ratio in R7s

Klu/WT1 has been shown to be a transcriptional repressor in other systems (*Drummond et al., 1992*; *McDonald et al., 2003*; *Kaspar et al., 2008*). As *sin* decreases Ss expression frequency and is predicted to increase Klu binding affinity, we hypothesized that Klu also represses stochastic *ss* expression in R7s. We found that Klu was expressed in R7s in larval eye imaginal discs in a Gaussian distribution (*Figure 4A–B*; *Figure 4—figure supplement 1*) (*Wildonger et al., 2005*). We predicted that increasing Klu levels would cause a decrease in the proportion of Ss$^{ON}$ R7s, whereas decreasing or completely ablating Klu would cause an increase in the proportion of Ss$^{ON}$ R7s. Indeed, increasing the levels of Klu in Klu-expressing cells (*klu > klu*), all photoreceptors (*eye > klu*), or specifically in all R7s (*R7 > klu*) caused a decrease in the proportion of Ss$^{ON}$ (Rh4) R7s (*Figure 4C–D*; *Figure 1—figure supplement 2C and E*). This decrease in the Ss$^{ON}$/Ss$^{OFF}$ ratio upon increasing Klu levels mimicked the effect of *sin*, consistent with *sin* increasing the binding affinity for the Klu repressor.

Conversely, *klu* loss-of-function mutants displayed increases in the proportion of Ss$^{ON}$ (Rh4) R7s (*Figure 4E–F*). We examined Ss expression directly and found that the proportion of Ss$^{ON}$ R7s increased in *klu* null mutants (*Figure 1—figure supplement 2D–E*). Moreover, we found that the proportion of Ss$^{ON}$ R7s increased in *klu* mutant clones compared to wild type clones (*Figure 4G–I*). As the proportion of Ss$^{ON}$ R7s increases specifically in *klu* mutant clones and decreases upon ectopic expression of Klu in R7s, we conclude that Klu is endogenously expressed at intermediate levels and acts cell-autonomously to determine Ss expression state.

Our data suggest that the ratio of Ss on/off gene expression is controlled by both the level of Klu protein and the binding affinity of the Klu site. To test this idea, we altered Klu levels in flies with the higher affinity Klu site (i.e. with *sin*). Because the proportion of Ss$^{ON}$ R7s is reduced in flies with

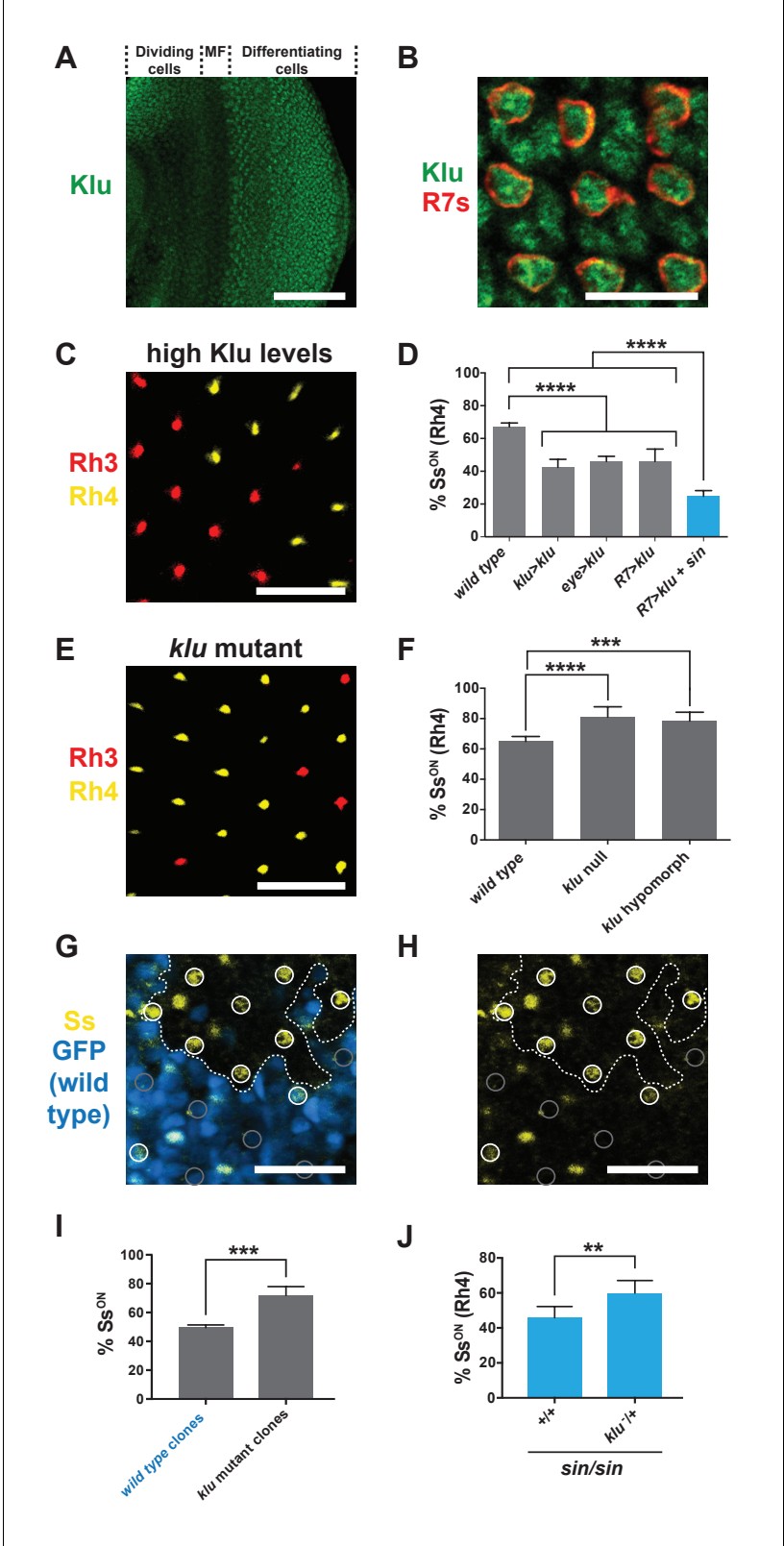

**Figure 4.** Levels of Klu determine the ratio of Ss^ON/Ss^OFF R7s. (A) Klu was expressed in the developing larval eye disc. MF indicates morphogenetic furrow. Scale bar indicates 100 μm. (B) Klu was expressed in all R7s in the developing larval eye disc. Red indicates R7s marked by *pm181 >Gal4, UAS > mCD8 GFP*. Scale bar indicates 10 μm. See also *Figure 4—figure supplement 1*. (C–D) Increasing Klu levels decreased the proportion of Ss^ON R7s. *Figure 4 continued on next page*

*Figure 4 continued*

Increasing Klu levels in flies with *sin* caused an additional reduction in the proportion of Ss$^{ON}$ R7s. In C, representative image of a retina from an *eye > klu* fly. R7s express either Rh3 (Ss$^{OFF}$) or Rh4 (Ss$^{ON}$). Scale bar indicates 20 μm. In D, **** indicates p<0.0001. Error bars indicate standard deviation (SD). (E–F) *klu* loss-of-function mutants displayed increases in the proportion of Ss$^{ON}$ R7s. In E, representative image of a retina from a *klu* hypomorph. R7s express either Rh3 (Ss$^{OFF}$) or Rh4 (Ss$^{ON}$). Scale bar indicates 20 μm. In F, **** indicates p<0.0001; *** indicates p<0.001. Error bars indicate standard deviation (SD). (G–H) The proportion of Ss$^{ON}$ R7s was higher in *klu* null mutant clones compared to wild type clones in mid-pupal retinas. GFP+ indicates wild type clone; GFP- indicates *klu* mutant clone. The dotted line marks the clone boundary. White circles indicate Ss$^{ON}$ R7s; gray circles indicate Ss$^{OFF}$ R7s. In addition to expression in circled R7s, Ss was expressed in bristle cells (unmarked). Scale bar indicates 20 μm. (I) Quantification of % Ss$^{ON}$ R7s in *klu* null mutant and wild type clones. *** indicates p<0.001. Error bars indicate standard deviation (SD). (J) Decreasing *klu* gene dosage in *klu* null mutant heterozygotes suppressed the *sin* phenotype. ** indicates p<0.01. Error bars indicate standard deviation (SD).
DOI: https://doi.org/10.7554/eLife.29593.012

The following figure supplement is available for figure 4:

**Figure supplement 1.** Klu is expressed in R7s in a Gaussian distribution.
DOI: https://doi.org/10.7554/eLife.29593.013

---

increased Klu levels (high repressor levels) or in flies with the *sin* variant (high binding affinity), we predicted a further reduction in flies with both high Klu and *sin* (high repressor levels, high binding affinity). We generated flies with increased levels of Klu in a *sin* genetic background and observed a significant additional reduction in the proportion of Ss$^{ON}$ R7s (*Figure 4D*).

To further test the relationship between Klu levels and binding site affinity, we reduced *klu* gene dosage in flies with *sin* and found that the *sin* phenotype was suppressed in *klu* mutant heterozygotes (*Figure 4J*). We conclude that *sin* increases Klu binding affinity and that the binding affinity of the Klu site and levels of Klu protein determine the proportion of Ss$^{ON}$ R7s.

## Discussion

Our studies of wild-derived flies revealed significant variation in stochastic Ss expression. We identified *sin*, a single base pair insertion in the ~60 kb *ss* locus that dramatically lowers the Ss$^{ON}$/Ss$^{OFF}$ ratio by increasing the binding affinity for the transcriptional repressor Klu. This decrease in Ss expression frequency changes the proportion of color-detecting photoreceptors and alters innate color preference in flies.

*sin* appears to be a relatively new mutation in *D. melanogaster* populations. *sin* is absent among diverse drosophilid species spanning millions of years of divergence (*Figure 3—figure supplement 1B–C*) and is segregating at an extremely low frequency among non-admixed African *D. melanogaster* lineages (*Figure 1—figure supplement 3A–F*). *sin* likely rose to intermediate frequencies following *D. melanogaster's* colonization of Europe about 10–15 thousand years ago (*Li and Stephan, 2006*). *sin* continues to segregate at intermediate frequencies amongst North American populations (*Figure 1—figure supplement 3A–F*), which were established within the last 150 years from mixtures of European and African populations (*Bergland et al., 2016*). The recent rise in the frequency of *sin* suggests that it could be the target of natural selection, perhaps via modulation of innate color preference. We tested this model by assessing patterns of allele frequency differentiation among populations sampled worldwide and by examining haplotype homozygosity surrounding *sin*. We compared these statistics at *sin* to the distribution of statistics calculated from several thousand randomly selected 1–2 bp indel polymorphisms that segregate at ~25% in the DGRP. Curiously, *sin* did not deviate from genome-wide patterns (*Figure 1—figure supplement 3G–J*) suggesting that it might be selectively neutral in contemporary *D. melanogaster* populations.

It is interesting that Rhodopsin expression varies so significantly in the wild, given the nearly invariant hexagonal lattice of ommatidia in the fly eye. Rhodopsins are G-protein coupled receptors (GPCRs), a class of proteins identified as a source of natural behavioral variation in worms, mice, and voles (*Young et al., 1999*; *Yalcin et al., 2004*; *Bendesky et al., 2011*). Dramatic differences in Rhodopsin expression patterns across insect species (*Hilbrant et al., 2014*; *Wernet et al., 2015*)

suggest that variation in the expression of GPCRs, rather than retinal morphology, may allow rapid evolution in response to environmental changes.

*sin* increases the binding affinity of a conserved Klu site, suggesting that the site is suboptimal or low-affinity for Klu binding. Low-affinity sites ensure the timing and specificity of gene expression (*Jiang and Levine, 1993*; *Gaudet and Mango, 2002*; *Scardigli et al., 2003*; *Rowan et al., 2010*; *Ramos and Barolo, 2013*; *Crocker et al., 2015*; *Farley et al., 2015*; *Crocker et al., 2016*). Our studies reveal a critical role for a low-affinity binding site in the regulation of a stochastically expressed gene. The suboptimal Klu site, bound by endogenous levels of Klu, yields the normal 65:35 $Ss^{ON}/Ss^{OFF}$ ratio. Changing the affinity of the site or the level of Klu alters the ratio of $Ss^{ON}/Ss^{OFF}$ cells. We conclude that stochastic on/off gene expression is controlled by threshold levels of *trans* factors binding to low-affinity sites.

The level of Klu (analog input) determines the binary on/off ratio of Ss expression (digital output). In contrast, gene regulation is best understood in cases where levels of transcription factors (analog input) regulate the levels of target gene expression (analog output). Interestingly, *sin* or genetic perturbation of *klu* affected the frequency of Ss expression (*Figures 1F, I–K* and *4C–J*, *Figure 1—figure supplement 2*) but not levels (*Figure 1—figure supplement 4*).

The on/off nature of Ss expression suggests a cooperative mechanism whereby Klu acts with other factors to regulate *ss.* Conservation of additional base pairs surrounding the Klu site (*Figure 3C*) is consistent with cooperative binding of Klu and others factors, possibly through dimerization or multimerization. These additional conserved base pairs could also enable binding of activating transcription factors to sites that overlap with the Klu site. These activating transcription factors may compete with the repressor Klu for binding to determine the stochastic on/off expression state of *ss.*

The expression state of *ss* could be determined by the intrinsic variation in Klu levels (*Figure 4—figure supplement 1*). In this model, if Klu levels exceed a threshold, *ss* is off, and if Klu levels are below the threshold, *ss* is on. Alternatively, Klu levels could set the threshold for a different gene regulatory mechanism, such as DNA looping or heterochromatin spreading. The regions encompassing and neighboring the Klu binding site drive gene expression in the eye (*Figure 1—figure supplement 1*), suggesting that complex interactions between this regulatory DNA element, the R7/R8 enhancer, and the two silencers (*Figure 1B*) ultimately control the *ss* on/off decision.

Cell fate specification is commonly thought of as a reproducible process whereby cell types uniformly express specific batteries of genes. This reproducibility is often the result of high levels of transcription factors binding to high-affinity sites, far exceeding a regulatory threshold, yielding expression of target genes in all cells of a given type. In contrast, the stochastic on/off expression of Ss requires finely tuned levels of regulators binding to low-affinity sites. We predict that fine tuning of binding site affinities and transcription factor levels will emerge as a common mechanistic feature that determines the ratio of alternative fates in stochastic systems.

## Materials and methods

### *Drosophila* genotypes and stocks

Flies were raised on standard cornmeal-molasses-agar medium and grown at 25°C.

| Short genotype | Complete genotype | Figures | Source |
|---|---|---|---|
| lab stock or wild type | *yw; sp/CyO;+* or *+ ; +; +* | 1C, 4D, 4F, *Figure 1—figure supplements 2A, E* and *4D–E* | |
| Low | $w^{1118}$/Y; DGRP-397 / +; DGRP-397/ Df(3R)Exel6269 | 1E | DGRP (*Mackay et al., 2012*) (*Parks et al., 2004*) |
| High | $w^{1118}$/Y; DGRP-229 / +; DGRP-229/ Df(3R)Exel6269 | 1E | DGRP (*Mackay et al., 2012*) |
| Wild-derived lines | $w^{1118}$/Y; DGRP / +; DGRP/Df(3R)Exel6269 | 1F, *Figure 1—source data 1* | DGRP (*Mackay et al., 2012*) |

*Continued on next page*

*Continued*

| Short genotype | Complete genotype | Figures | Source |
|---|---|---|---|
| African Line No *sin* (AL-1) | *KN34; KN34; KN34* | 1I | (*Lack et al., 2015*) |
| African Line No *sin* (AL-2) | *KR42; KR42; KR42* | 1I | (*Lack et al., 2015*) |
| African Line *sin* (AL-3) | *KN133N; KN133N; KN133N* | 1I | (*Lack et al., 2015*) |
| African Line *sin* (AL-4) | *KR7; KR7; KR7* | 1I | (*Lack et al., 2015*) |
| Lab Line No *sin* (LL-1) | *yw; +; +/Df(3R)Exel6269* | 1J | |
| Lab Line *sin* (LL-2) or *sin/ss df* | *yw; +; TM6B (with sin)/Df(3R)Exel6269* | 1J, **Figure 1—figure supplement 4E** | |
| CRISPR No *sin*(CL-1) | *yw; +; CRISPR negative/Df(3R)Exel6269* | 1K | |
| CRISPR *sin* (CL-2) | *yw; +; CRISPR positive 1/Df(3R)Exel6269* | 1K | |
| CRISPR *sin* (CL-3) | *yw; +; CRISPR positive 2/Df(3R)Exel6269* | 1K | |
| CRISPR *sin* (CL-4) | *yw; +; CRISPR positive 3/Df(3R)Exel6269* | 1K | |
| Innate color preference | *DGRP; DGRP; DGRP* | 2, **Figure 2—source data 1** | DGRP (*Mackay et al., 2012*) |
| Wild type (control for optimal Klu site) | *yw; +; +/Df(3R)Exel6269* | 3E | |
| CRISPR Optimal Klu site 1 | *yw; +; CRISPR positive 1/Df(3R)Exel6269* | 3E | |
| CRISPR Optimal Klu site 2 | *yw; +; CRISPR positive 2/Df(3R)Exel6269* | 3E | |
| Klu expression in larval eye imaginal discs | *yw; pm181 > Gal4, UAS > mCD8 GFP/CyO; +* | 4A, 4B | |
| *klu > klu* | *yw; +; klu > Gal4/UAS > klu* | 4D | (*Klein and Campos-Ortega, 1997*) |
| *eye > klu* | *yw; +; lGMR > Gal4/UAS > klu* | 4C, 4D, **Figure 1—figure supplement 2C and E**, 4E | (*Wernet et al., 2003*) |
| *R7 > klu* | *yw; panR7 > Gal4 / +; UAS > klu/+* | 4D | (*Chou et al., 1999b1999*) |
| Wild type (control for *klu* mutants) | *yw; ey > Gal4, UAS > flp/+; GMR > hid, cL FRT2A/ubi > GFP FRT2A* | 4F | |
| *klu* null | *yw; ey > Gal4, UAS > flp/+; GMR > hid,cL FRT2A/kluR$^{R51}$ FRT2A* | 4F, **Figure 1—figure supplement 2D–E**, **Figure 1—figure supplement 4E** | (*Klein and Campos-Ortega, 1997*; *Stowers and Schwarz, 1999*) |
| *klu* hypomorph | *yw; ey > Gal4, UAS > flp/+; GMR > hid, cL FRT2A/kluPBac$^{PBac\ LL02701}$ FRT2A FRT82b P[Car20y]* | 4E, 4F | (*Stowers and Schwarz, 1999*; *Schuldiner et al., 2008*) |
| *klu* null mutant clones | *ey > flp; +/CyO; klu$^{R51}$ FRT2A/ ubi > GFP FRT2A* | 4G-I | |
| *+/+, sin/sin* | *+; +; sin* | 4J, **Figure 1—figure supplement 2B and E** | |
| *klu⁻/+, sin/sin* | *+; +; klu$^{R51}$ FRT2A, sin / +, sin* | 4J | |
| *R7 reporter* | *yw; pm181 > Gal4, UAS > mCD8 GFP/CyO; +* | **Figure 4—figure supplement 1** | |
| 14 F02 | *w$^{1118}$; P{GMR14F02-GAL4}attP2 / +; UAS > H2B:YFP / +* | **Figure 1—figure supplement 1A–B** | (*Jenett et al., 2012*) |
| 15 F08 | *w$^{1118}$; P{GMR15F08-GAL4}attP2 / +; UAS > H2B:YFP / +* | **Figure 1—figure supplement 1A and C** | (*Jenett et al., 2012*) |
| 15 F02 | *w$^{1118}$; P{GMR15F02-GAL4}attP2 / +; UAS > H2B:YFP / +* | **Figure 1—figure supplement 1A and D** | (*Jenett et al., 2012*) |

## Antibodies

Antibodies were used at the following dilutions: mouse anti-Rh3 (1:100) (gift from S. Britt, University of Colorado), rabbit anti-Rh4 (1:100) (gift from C. Zuker, Columbia University), rat anti-Klu (1:200) (gift from C. Desplan, New York University), guinea pig anti-Ss (1:500) (gift from Y.N. Jan, University of California, San Francisco), sheep anti-GFP (1:500) (BioRad, Hercules, CA, USA), rat anti-ELAV (1:50) (DSHB, Iowa City, Iowa, USA), mouse anti-Pros (1:10) (DSHB), and Alexa 488 Phalloidin (1:80) (Invitrogen, Thermo Fisher Scientific, Waltham, MA, USA). All secondary antibodies were Alexa Fluor-conjugated (1:400) and made in donkey (Molecular Probes).

## Antibody staining

Adult, mid-pupal, and larval retinas were dissected as described (*Hsiao et al., 2012*) and fixed for 15 min with 4% formaldehyde at room temperature. Retinas were rinsed three times in PBS plus 0.3% Triton X-100 (PBX) and washed in PBX for >2 hr. Retinas were incubated with primary antibodies diluted in PBX overnight at room temperature and then rinsed three times in PBX and washed in PBX for >4 hr. Retinas were incubated with secondary antibodies diluted in PBX overnight at room temperature and then rinsed three times in PBX and washed in PBX for >2 hr. Retinas were mounted in SlowFade Gold Antifade Reagent (Invitrogen). Images were acquired using a Zeiss LSM 700 confocal microscope.

## Quantification of expression

Frequency of Rh3 (Ss$^{OFF}$) and Rh4 (Ss$^{ON}$) expression in R7s was scored in adults. Six or more retinas were scored for each genotype (N). 100 or more R7s were scored for each retina (n). Frequency was assessed using custom semi-automated software (see below) or manually.

Frequency of Ss expression in R7s was assessed with a Ss antibody in mid-pupal animals. Four or more retinas were scored for each genotype (N). 70 or more R7s were scored for each retina (n). Frequency was assessed manually.

Levels of Ss expression in Ss$^{ON}$ R7s were assessed with a Ss antibody in mid-pupal animals. Three retinas were scored (N). 40 or more Ss$^{ON}$ R7s were scored for each retina (n). We used ImageJ software to quantify Ss levels in Ss$^{ON}$ R7s (*Figure 1—figure supplement 4A–C*). A circular ''region of interest'' was manually placed at the center of each Ss$^{ON}$ R7 (identified by expression of Ss and the R7 marker Prospero) to avoid signal from neighboring photoreceptors. ImageJ software assessed the mean pixel intensity for each region of interest for each Ss$^{ON}$ R7.

Levels of Klu expression in R7s were assessed with a Klu antibody in third instar larval animals. Five retinas were scored (N). 65 or more R7s were scored for each retina (n). We used ImageJ software to quantify Klu levels in all R7s (*Figure 4—figure supplement 1*). A circular ''region of interest'' was manually placed at the center of each R7 (identified by *pm181 >GAL4; UAS > GFP* reporter expression) to avoid signal from neighboring photoreceptors. ImageJ software assessed the mean pixel intensity for each region of interest for each R7.

To determine the number of rows from the equator to the dorsal third region of the adult retina, we first used phalloidin to stain actin (marking rhabdomeres of ommatidia) to locate the equator of each retina. We then counted the number of rows from the equator to the first R7 cell with co-expression of Rh3 and Rh4.

## Image processing

We employed a custom algorithm to identify the positions of individual R7 photoreceptors within an image of the fly retina. First, individual fluorescence images from each wavelength channel were denoised using a homomorphic filter (*Oppenheim et al., 1968*) and Gaussian blur. Next, R7 boundaries were located using the Canny edge detection method (*Canny, 1986*). Cells were then roughly segmented using the convex hull algorithm (*Barber et al., 1996*). Active contouring (*Chan and Vese, 2001*) was used to refine the segments to fit the R7s more closely. Finally, a watershed transform was applied to the image, dividing it into regions that each contain a single R7. Regions were excluded by size or distance from the center to prevent artifacts due to the curvature of the fly retina. For the remaining regions, normalized intensities from the Rh3 and Rh4 channels were compared in order to assign each region a label, indicating that its R7# is stained with Rh3 or Rh4. A

MATLAB (The MathWorks, Inc.) script that implements our algorithm is available at https://app.assembla.com/spaces/roberts-lab-public/wiki/Fly_Retina_Analysis.

## Genome-Wide association studies

Genotype data from the DGRP freeze two lifted to the dm6/BDGP6 release of the *D. melanogaster* genome was obtained from (ftp://ftp.hgsc.bcm.edu/DGRP/). Phenotypes were calculated for the progeny of crosses of DGRP lines and *Df(3R)Exel6269* flies. To estimate genotypes of these flies from the DGRP data, we simulated each cross. For each SNP or indel variant in the DGRP genotype data, we assigned a new genotype: (1) homozygous reference remains homozygous reference, (2) homozygous alternate maps to homozygous alternate *if* in deficiency region, otherwise heterozygous, and (3) all other genotypes mapped to missing or unknown and not included in subsequent analyses. We performed quantitative trait association analysis using plink2 –linear (version 1.90 beta 25 Mar 2016; PMID:25722852). To reduce the impact of population structure, we included the first 20 principal components of the standardized genetic relationship matrix as covariates (calculated using plink2 –pca). To empirically correct p-values for each site, we performed a max(T) permutation test with 10,000 permutations (mperm option to plink2).

## CRISPR-mediated mutagenesis

*sin* was inserted into a lab stock line using CRISPR (*Gratz et al., 2013*; *Port et al., 2014*). Sense and antisense DNA oligos for the forward and reverse strands of the gRNA were designed to generate BbsI restriction site overhangs. The oligos were annealed and cloned into the pCFD3 cloning vector (Addgene, Cambridge, MA). A single stranded DNA homology bridge was generated with 60 bp homologous regions flanking each side of the predicted cleavage site. The gRNA construct (500 ng/ul) and homology bridge oligo (100 ng/ul) were injected into *Drosophila* embryos (BestGene, Inc.). Single males were crossed with a balancer stock (*yw; +; TM2/TM6B*), and F1 female progeny were screened for the insertion via PCR and sequencing. Single F1 males whose siblings were *sin*-positive were crossed to the balancer stock (*yw; +; TM2/TM6B*) and the F2 progeny were screened for the insertion via PCR and sequencing. *sin*-negative flies from a single founder were used to establish a stable stock (CL-1) and *sin*-positive flies from three founders were used to establish independent stable stocks (CL-2, CL-3, CL-4).

| Oligo name | Sequence |
| --- | --- |
| Homologous bridge | TCTCTCTCTCTCTGTGTGTGTGTCACTCACAAATGACAACGTGGTGTGGGCGTCGAATATATGAGTTACTTCGCACCCAGCCAGCCAAGCCAGAGCAAATTGAGCCAAACCAAAGCAAA |
| gRNA F | GTCGTGTGTGTGGGCGTCGAATATAT |
| gRNA R | AAACATATATTCGACGCCCACACA |
| Genotype F | GCCACCCTTCGACCATTTTGG |
| Genotype R | GTCAGCCACTACATGGTTTCG |

The Klu optimal site was generated in a lab stock line using CRISPR (*Gratz et al., 2013*; *Port et al., 2014*). CRISPR was performed with the same gRNA and genotyping primers as described above, but with a new homologous bridge donor.

| Oligo name | Sequence |
| --- | --- |
| Homologous bridge | TCTCTCTCTCTCTGTGTGTGTGTCACTCACAAATGACAACGTGCGTGGGCGTCGAATATATGAGTTACTTCGCACCCAGCCAGCCAAGCCAGAGCAAATTGAGCCAAACCAAAGCAAA |
| gRNA F | GTCGTGTGTGGGCGTCGAATATAT |
| gRNA R | AAACATATATTCGACGCCCACACA |
| Genotype F | GCCACCCTTCGACCATTTTGG |
| Genotype R | GTCAGCCACTACATGGTTTCG |

## T-maze behavioral assays

Adult flies were raised on standard medium on a 14 hr/10 hr light and dark cycle at 25°C. The behavioral assay room was illuminated by a 630 nm red LED bulb (superbrightleds.com; PAR30IP-x8-90) whose emitted light lies outside of the sensitivity spectrum for fly photodetection. For each trial, 100 female flies were starved for 8 hr and then inserted into the elevator of the T-maze (Robert Eifert, Bayshore, NY). The elevator was lowered to a junction that, on each side, held an unused plastic tube (Falcon 352017). The T-maze was covered in black chalkboard tape and the plastic tubes were painted black. The T-maze and lights were kept a constant distance apart by a custom 3D-printed holder. A blue LED light (450 nm) and a green LED light (525 nm) on opposite sides were simultaneously turned on. Blue and green LED lights were obtained from superbrightleds.com (E12-B5, E12G5). The blue light was covered with three layers of 3x neutral density (ND) filters, while the green light was covered with one layer. After 20s, the lights were turned off and the tubes were removed and capped. Flies from each tube were counted and the preference index (PI) was calculated using the formula $PI = (N_G - N_b) / (N_G + N_b)$, where $N_G$ equals the number of flies in the tube illuminated with green light and $N_b$ equals the number of flies in the tube illuminated with blue light. PI ranged from $-1$ to 1, with negative values indicating a blue preference and positive values indicating a green preference. Five or more trials were conducted for each genotype (N). 100 or more flies were scored for each trial (n).

## Consensus sequence

For the B1H data sets, WebLogo3 was used to generate position weight matrices (PWMs) (*Zhu et al., 2011*; *Enuameh et al., 2013*)(*Figure 3—figure supplement 1A*). For the SELEX-SEQ data sets, MEME-ChIP version 4.11.2 was used to generate PWMs (*Machanick and Bailey, 2011*; *Nitta et al., 2015*) (ENA: ERX606541-ERX606544). Motif discovery and enrichment mode was set to normal, and the 1st order model of sequences was used as the background model. Expected motif site distribution was set at zero or one occurrence per sequence. Minimum width of motifs to be found by MEME was set to 12, and the max was set to 20.

## Conservation analysis

The Klu site and neighboring sequences for 21 *Drosophila* species were obtained from the UCSC genome browser. TOMTOM version 4.11.2. was used to generate the conservation PWM (*Gupta et al., 2007*)(*Figure 3C*, *Figure 3—figure supplement 1B*).

## SELEX-seq analysis

SELEX-seq datasets from (*Nitta et al., 2015*) were obtained from ENA (ERX606541-ERX606544). For read-level analysis, we counted the number of reads containing the Klu binding site with *sin*, without *sin*, and neither site (there were no reads with both sites). We performed McNemar's test to assess significance. We computed the frequency of each 10-mer within each dataset using Jellyfish version 2.2.6 (*Marçais and Kingsford, 2011*). Using these counts, we determined the number of 10-mers with frequency greater than that of the Klu binding site with and without *sin*. Frequencies reported are for the combination of all four SELEX datasets.

Jellyfish 2.2.6 was used to canonically count kmers of length 10 with an initial hash of size 100M from fasta files generated from the first and fourth rounds of selection. Kmers were reverse complemented as necessary to minimize the hamming distance from the consensus sequence. Reported counts come from the fourth round.

## Population genetic analyses

We estimated allele frequencies from populations sampled worldwide at *sin* and at other 1–2 bp indel polymorphisms. Allele frequency estimates based on pooled resequencing of populations sampled in North America and Europe were obtained from (*Bergland et al., 2014*) and (*Kapun et al., 2016*). Allele frequencies based on haplotypes (*Lack et al., 2016*) were also obtained from populations sampled in North America, the Caribbean, Europe, and Africa.

For pooled samples, we mapped raw sequence reads to Release 6 of the *Drosophila melanogaster* genome, removed PCR duplicates, performed indel-realignment using GATK version, and called allele frequencies using VarScan. For haplotype data, we relied on published indel VCF files

obtained from the Drosophila Genome Nexus (DGN; http://www.johnpool.net/genomes.html). Regions of admixture in African genomes were identified based on analyses by *Lack et al., 2016*. DGN data were mapped to Release 5 of the *Drosophila* genome and we converted those coordinates to Release six using the lift-over file available from the UCSC Genome browser (http://hgdownload.soe.ucsc.edu/goldenPath/dm3/liftOver/dm3ToDm6.over.chain.gz).

We sought to assess whether the distribution of *sin* among populations sampled worldwide was significantly different than expected by chance based on other comparable indel polymorphisms. *sin* was originally identified in the Drosophila Genetic Reference Panel (DGRP), derived from a population in Raleigh, NC, where it segregates at ~25%. We observed that *sin* segregates at ~10–25% in other North American and European populations but is rare/absent in ancestral African populations (*Figure 1—figure supplement 3A–F*). Such changes in allele frequency among continents could indicate the action of positive selection. To test this model, we identified ~1500 other 1–2 bp autosomal indel polymorphisms that segregate at $25 \pm 5\%$ in the DGRP (hereafter, 'control set'). We estimated $F_{ST}$ among continents as well as within North America at *sin* and our control set. $F_{ST}$ values were rank normalized and converted to a Z-score through an inverse normal CDF with mean zero and standard deviation one. *sin* did not show elevated levels of $F_{ST}$ within or between continents relative to the control set, suggesting that *sin* does not contribute to local adaptation amongst sampled populations (*Figure 1—figure supplement 3G–H*).

Next, we tested whether haplotype patterns surrounding *sin* are indicative of a partial selective sweep. We calculated the extended haplotype homozogosity (EHH) score and integrated EHH (iEHH) score for haplotypes with and without *sin* (derived and ancestral haplotypes, respectively) in the DGRP data where *sin* was originally identified. EHH scores were also calculated at the control set, as described above. EHH scores were calculated using the R package *rehh* (*Gautier and Vitalis, 2012*). The derived *sin* allele shows an elevated iEHH score compared to the ancestral allele, suggestive of a partial selective sweep (*Figure 1—figure supplement 3I–J*). To test this model, we calculated the integrated haplotype statistic as,

$$\mathrm{IHS} = \log2\left(\mathrm{iEHH}_{\mathrm{derived}} / \mathrm{iEHH}_{\mathrm{ancestral}}\right)$$

for *sin* as well as the control set. IHS for *sin* is not significantly different than expected by chance relative to other comparable indel polymorphisms (*Figure 1—figure supplement 3I–J*).

## Acknowledgements

We are grateful to Steve Britt, Claude Desplan, Lily Jan, Yuh-Nung Jan, Thomas Klein, Chuck Langley, John Pool, Charles Zuker, and the Bloomington and Kyoto Stock Centers for generously providing published fly stocks and antibodies. We thank Erik Anderson, Greg Bashaw, John Kim, Alex Kolodkin, Rejji Kuruvilla, Chris Potter, Geraldine Seydoux, Mark Van Doren and members of the Johnston lab for helpful comments on the manuscript. RJJ was supported by the Pew Scholar Award 00027373 and NIH R01EY025598.

## Additional information

### Funding

| Funder | Grant reference number | Author |
| --- | --- | --- |
| National Eye Institute | R01EY025598 | Robert J Johnston Jr |
| Pew Charitable Trusts | 00027373 | Robert J Johnston Jr |

The funders had no role in study design, data collection and interpretation, or the decision to submit the work for publication.

### Author contributions

Caitlin Anderson, Conceptualization, Data curation, Supervision, Funding acquisition, Investigation, Visualization, Methodology, Writing—original draft, Project administration, Writing—review and editing; India Reiss, Conceptualization, Data curation, Investigation, Visualization, Methodology,

Writing—original draft, Writing—review and editing; Cyrus Zhou, Conceptualization, Data curation, Investigation, Methodology; Annie Cho, Haziq Siddiqi, Benjamin Mormann, Investigation, Methodology; Cameron M Avelis, Software, Investigation, Methodology; Peter Deford, Data curation, Software; Alan Bergland, Elijah Roberts, Resources, Data curation, Software, Formal analysis, Writing—review and editing; James Taylor, Daniel Vasiliauskas, Data curation, Software, Formal analysis, Investigation, Methodology; Robert J Johnston, Conceptualization, Data curation, Supervision, Funding acquisition, Investigation, Methodology, Writing—original draft, Project administration, Writing—review and editing

### Author ORCIDs
Caitlin Anderson http://orcid.org/0000-0003-2260-6209
Alan Bergland http://orcid.org/0000-0001-7145-7575
James Taylor http://orcid.org/0000-0001-5079-840X
Robert J Johnston http://orcid.org/0000-0002-5775-6218

### Decision letter and Author response
Decision letter https://doi.org/10.7554/eLife.29593.016
Author response https://doi.org/10.7554/eLife.29593.017

# Additional files

### Supplementary files
• Transparent reporting form
DOI: https://doi.org/10.7554/eLife.29593.014

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
