## [Decision Letter]

Thank you for submitting your article "Natural variation in stochastic photoreceptor specification and color preference in *Drosophila*" for consideration by *eLife*. Your article has been reviewed by two peer reviewers, and the evaluation has been overseen by Simon Sprecher as Guest Reviewing Editor and Patricia Wittkopp as the Senior Editor. The following individual involved in review of your submission has agreed to reveal his identity: Stein Aerts (Reviewer #3).

The reviewers have discussed the reviews with one another and the Reviewing Editor has drafted this decision to help you prepare a revised submission.

Summary:

The study addresses how stochastic gene expression may be modulated by investigating the impact of a naturally occurring variation in the spineless locus. Identification of a single base pair insertion that may change the binding of the transcription factor Klumpfuss. The presence of the *sin* variance shifts the ratio two ommatida types as well as the behavioural preference between blue and green light. These findings provide insight into how inter-individual differences in nervous system architecture and behaviour may arise.

Essential revisions:

While the manuscript was perceived positively there are however a few points that should be addressed. As you will see from the points depicted below a main point from several sides concerns the putative mechanistic link between *klu*, the *sin* locus and Ss expression. More direct evidence for the altered expression of Ss by the *sin* locus or *klu* binding to variants of the *sin* locus would help to resolve this criticism.

1) The hypothesis drawn by the authors is that Klu represses Ss expression in R7 cells. It would be helpful to examine expression levels of Ss in R7 cells in Klu mutants and animals with increased Klu.

2) The authors claimed that Klu was expressed in R7 cells of the larval eye disc at 'intermediate levels' and altering Klu levels changed the SsON/SsOFF ratio. Can this intermediate level of Klu be quantified? It appears in Figure 4 that endogenous *klu* is differentially expressed in R7 cells.

3) The authors used bioinformatics to analyze available SELEX-seq binding data sets from Nitta, et al. 2015 and concluded that *sin* increased the binding affinity for Klu based on the enrichment of the number of reads obtained with *sin*. Since this was solely based on in vitro data on a randomized set of DNA with N-terminal thioredoxin-HIS tag fusion proteins, a more thorough characterization is necessary to confirm this claim. Would the same tagged-Klu expressed in vivo rescue SsON/SsOFF photoreceptor ratio in Klu null? The authors should further demonstrate altered DNA binding affinity for Klu using a specific DNA probe with or without *sin* in EMSA assays.

4) The insertion of a C does not seem to affect the core Klu binding site, but rather a flanking nucleotide, and this binding site is a CA rich stretch where an extra C is added (CGCCCACACA to CGCCCACACCA). The authors argue that this may change the affinity of Klu binding. Given that Klu is expressed in the R7, and that perturbations of Klu phenocopy *sin*, this is a plausible explanation. Nevertheless, no experimental validation is provided to show that Klu actually binds to this region and that the binding is affected by the insertion. I realise that ChIP-seq is difficult given the limited amount of material. Can another experiment be thought of? If not, can this be argued in the text – that this evidence is indirect, and no formal proof is provided that Klu binding in vivo is affected by this insertion. The authors provide surrogate evidence that Klu SELEX-seq is biased towards CGCCCACACCA sequences having *sin*; this is interesting but it is not convincing to me, because the SELEX logo's do not contain this flanking site, rather they suggest Klu binds to CGCCCACGCA. This is quite different from the site with *sin* (notice the trailing GCA). Given the very high conservation of a 17 bp stretch that contains a match to Klu, an alternative hypothesis is that actually two (or even more) transcription factors may bind to this 17bp stretch, perhaps Klu together with another factor, and that *sin* affects binding of the other, yet unknown factor, rather than of Klumpfuss. Klu perturbations would still affect Ss expression in that scenario, but not because of differential affinity with the site, but simply because Klu regulates Ss (in other words, Klu perturbations are not performed with a *sin* comparison, so they merely confirm that Klu regulates Ss). A possible experiment would be to perform the Klu perturbations with and without the *sin* insertion. Or to clone this region near an enhancer-reporter construct that activates a reporter, for example, in S2 cells (a strong enhancer that is active in S2 cells, many are known). Given the prediction of Klu-mediated repression, it can be investigated whether the reporter can be repressed by the *sin*-encompassing region, when transfecting with Klu cDNA, and whether this effect is changed between the wild type and the *sin* containing sequence. in vivo this would obviously be better, but would require more time. I am aware such experiments are challenging, so other experiments can be proposed by the authors, or if they are not possible given the limited time, the lack of formal evidence what this insertion does, and alternative hypotheses, should be discussed thoroughly (perhaps with a figure/cartoon).

5) Given that the Klu binding site is not very informative (lots of C's), how significant is the prediction that this is a Klu binding site? On 100 random genes with similar size as Ss, how many Klu matches are found? Can this be reported? Does the score change when *sin* is included? That is, does the Klu PWM score increase with *sin* compared to control/null sites – likely not because *sin* is not present in the PWM, correct? Can this be discussed in the paper. This is important because the PWM is a measure for the binding affinity, which is argued to be increased with *sin*, so the PWM score should change (significantly?).

6) It is written "As *sin* increases Klu binding affinity" – can this be changed into "As *sin* is predicted to increase Klu binding affinity"?

7) The role of the encompassing genomic region, how it is involved in the regulation of Ss, is not investigated – is this a new regulatory element of Ss? Are there any Janelia Flylight or VDRC tiles overlapping this region? If not, please discuss this. If there is time to clone this region into an enhancer-reporter and create a transgenic fly, that would be informative. If Flylight/VDRC lines are available, can they be tested with UAS-GFP to investigate whether this region is actually an Ss enhancer.

---

## [Author Response]

Essential revisions:While the manuscript was perceived positively there are however a few points that should be addressed. As you will see from the points depicted below a main point from several sides concerns the putative mechanistic link between klu, the sin locus and Ss expression. More direct evidence for the altered expression of Ss by the sin locus or klu binding to variants of the sin locus would help to resolve this criticism.1) The hypothesis drawn by the authors is that Klu represses Ss expression in R7 cells. It would be helpful to examine expression levels of Ss in R7 cells in Klu mutants and animals with increased Klu.

To address this concern, we used a Ss antibody to directly evaluate both the levels and on/off ratio of Ss expression in different genetic conditions. The results of the experiments described below are consistent with the conclusion that *sin* and Klu affect frequency but not levels of Ss expression.

Specifically, we performed three sets of experiments:

1) Direct analysis of Ss levels in wild type and *klu* mutant Ss^ON^ R7s: As comparison between retinas from different animals can complicate expression analysis, we generated *klu* null mutant clones. We examined Ss expression directly using a Ss antibody and found that Ss expression levels were not significantly different between wild type and *klu* mutant Ss^ON^R7s (Figure 1—figure supplement 4). These data indicate that Klu does not regulate Ss expression levels.

2) Analysis of retinal regions of Rhodopsin exclusivity and co-expression, as a proxy for Ss expression levels, in *klu* loss- and gain-of-function *sin* backgrounds and in flies with *sin*:We previously showed that changes in Ss levels or activity alter the exclusivity or co-expression of Rhodopsin expression (Thanawala et al. 2013). The main region of the retina is a random mosaic of R7 cells with exclusive expression of Rh3 (Ss^OFF^) or Rh4 (Ss^ON^). This exclusivity breaks down in the dorsal third region of the retina where some R7s exclusively express Rh3 (Ss^OFF^), whereas others co-express Rh4 and Rh3 (Ss^ON^). This dorsal third region starts approximately four rows above the dorsal/ventral equator of the retina (Figure 1—figure supplement 4). Increasing levels of Ss in Ss^ON^ R7s generates retinas with exclusive expression of Rh3 or Rh4 throughout the retina. Decreasing levels of Ss leads to an expansion of the dorsal third region of co-expression. Thus, we used the demarcation of the region of co-expression of Rh4 and Rh3 in Ss^ON^ R7s as a proxy for Ss levels. We determined the first row of dorsal third co-expression above the equator in *sin* hemizygotes, *klu* null mutants, and flies with Klu ectopic expression (*eye>klu*) and found no significant differences from wild type (Figure 1—figure supplement 4). Since these same genotypes cause changes in Ss expression frequency (Figure 1, Figure 4), we conclude that these genetic perturbations affect Ss expression frequency but not levels.

3) Direct analysis of Ss^ON^/Ss^OFF^ ratio in flies with *sin,* flies with increased Klu protein levels, and *klu* null mutants:We directly examined the ON/OFF ratio of Ss expression in R7s using a Spineless antibody. We found that *sin* and increased Klu protein levels caused a decrease in the proportion of Ss^ON^ R7s (Figure 1—figure supplement 2, and E), whereas *klu* null mutants displayed an increase (Figure 1—figure supplement 2). These findings are consistent with our observations using Rh4 as a marker for Ss^ON^ fate (Figure 1, Figure 4) and with our previous observations that Rh4 faithfully reports Ss expression in R7s (Johnston and Desplan 2014).

Additionally, we found that the proportion of Ss^ON^ R7s increased in *klu* mutant clones compared to wild type clones (Figure 4). Together with the decrease in the proportion of Ss^ON^ R7s observed upon ectopic expression of Klu in R7s (Figure 4), these data suggest an autonomous role for Klu-mediated regulation of *ss* in R7s.

The following additions and/or changes to the text describe these experiments:

“Interestingly, *sin* or genetic perturbation of *klu* affected the frequency of Ss expression (Figure 1, Figure 4, Figure 1—figure supplement 2) but not levels (Figure 1—figure supplement 4).”

“Figure 1—figure supplement 4. *klu* and *sin* genetic perturbations do not affect levels of Ss expression in Ss^ON^ R7s […] Quantification of the position of the dorsal third in number of rows above the equator. ns indicates not significant, p>0.05. Error bars indicate standard deviation (SD).”

“Using a Ss antibody, we examined Ss expression directly and found that flies homozygous for CRISPR *sin* alleles displayed a significant decrease in the proportion of Ss^ON^ R7s (Figure 1—figure supplement 2, and E).”

“Indeed, increasing the levels of Klu in Klu-expressing cells (*klu>klu*), all photoreceptors (*eye>klu*), or specifically in all R7s (*R7>klu*) caused a decrease in the proportion of Ss^ON^ (Rh4) R7s (Figure 4; Figure 1—figure supplement 2).”

“We examined Ss expression directly and found that the proportion of Ss^ON^ R7s increased in *klu* null mutants (Figure 1—figure supplement 2).”

“Figure 1—figure supplement 2. *sin* and *klu* genetic perturbations alter the proportion of Ss^ON^ R7s […] E. Quantification of A-D. The proportion of Ss^ON^ R7s decreased in flies with *sin* or ectopic expression of Klu and increased in *klu* null mutants. ** indicates p<0.01; * indicates p<0.05. Error bars indicate standard deviation (SD).”

“Moreover, we found that the proportion of Ss^ON^ R7s increased in *klu* mutant clones compared to wild type clones (Figure 4). As the proportion of Ss^ON^ R7s increases specifically in *klu* mutant clones and decreases upon ectopic expression of Klu in R7s, we conclude that Klu is endogenously expressed at intermediate levels and acts cell-autonomously to determine Ss expression state.”

“Figure 4. Levels of Klu determine the ratio of Ss^ON^/Ss^OFF^ R7s […] I. Quantification of% Ss^ON^ R7s in *klu* null mutant and wild type clones. *** indicates p<0.001. Error bars indicate standard deviation (SD).”

2) The authors claimed that Klu was expressed in R7 cells of the larval eye disc at 'intermediate levels' and altering Klu levels changed the SsON/SsOFF ratio. Can this intermediate level of Klu be quantified? It appears in Figure 4 that endogenous klu is differentially expressed in R7 cells.

To address the reviewer’s concern, we have quantified Klu expression in R7s in the larval eye. Klu levels were observed in a normal distribution, consistent with our statement that Klu is expressed at intermediate levels and that altering Klu levels changes the Ss^ON^/Ss^OFF^ ratio (Figure 4—figure supplement 1). Together with the observations that raising Klu levels decreased the Ss^ON^/Ss^OFF^ ratio and ablating *klu* increased the ratio, these data suggest that Klu is endogenously expressed at intermediate levels to determine the proportion of Ss-expressing R7s.

“We found that Klu was expressed in R7s in larval eye imaginal discs in a Gaussian distribution (Figure 4; Figure 4—figure supplement 1) (Wildonger et al. 2005).”

“Figure 4—figure supplement 1. Klu is expressed in R7s in a Gaussian distribution. A. Quantification of Klu levels in a single retina. B. Quantification of Klu levels across several retinas (n=5).”

3) The authors used bioinformatics to analyze available SELEX-seq binding data sets from Nitta, et al. 2015 and concluded that sin increased the binding affinity for Klu based on the enrichment of the number of reads obtained with sin. Since this was solely based on in vitro data on a randomized set of DNA with N-terminal thioredoxin-HIS tag fusion proteins, a more thorough characterization is necessary to confirm this claim. Would the same tagged-Klu expressed in vivo rescue SsON/SsOFF photoreceptor ratio in Klu null? The authors should further demonstrate altered DNA binding affinity for Klu using a specific DNA probe with or without sin in EMSA assays.

Mutation of the endogenous Klu site to a predicted high affinity site phenocopies *sin:* Since *sin* is predicted to increase the binding affinity for Klu and since *sin* caused a reduction in the on/off ratio of Ss expression, we hypothesized that making the Klu site a predicted optimal site (based on the PWM) would also cause a decrease in Ss^ON^ R7s. We used CRISPR to mutate the endogenous low affinity Klu site (ACGCCCAC**A**CAC) to the predicted high affinity site (ACGCCCAC**G**CAC) and observed a similar decrease in the proportion of Ss^ON^ R7s as in flies with *sin* (Figure 3). This result, in which an optimal high affinity Klu binding site phenocopies *sin,* strongly argues that the phenotype observed in flies with *sin* is due to an increase in binding affinity for the Klu transcription factor.

“Since *sin* is predicted to increase the binding affinity for Klu and *sin* caused a reduction in the on/off ratio of Ss expression, we hypothesized that mutating the Klu site to an optimized high-affinity site would also cause a decrease in the proportion of Ss^ON^ R7s. […] The observation that an optimized high-affinity Klu site causes a similar phenotype as *sin* is consistent with the conclusion that *sin* increases the binding affinity for Klu.”

“Figure 3. sin increases the binding affinity for the transcription factor Klumpfuss […] Conservation logo of the Klu site and neighboring sequence in the ss locus. Height of bases indicates degree of conservation. See also Figure 3—figure supplement 1.”

ChIP-qPCR analysis using the Klu antibody: In an effort to evaluate changes in Klu binding caused by *sin* in vivo, we conducted extensive ChIP-qPCR experiments using the Klu antibody and primers flanking the Klu site with or without *sin*. Though our data point toward enrichment in Klu binding in flies with *sin*, our results are not publication-quality due to high background binding in negative control experiments. In the future, we will use CRISPR to tag the endogenous Klu with GFP and use a commercially available ChIP-grade GFP antibody for the immunoprecipitation, however these experiments are not practical in the timeframe of this current work.

4) The insertion of a C does not seem to affect the core Klu binding site, but rather a flanking nucleotide, and this binding site is a CA rich stretch where an extra C is added (CGCCCACACA to CGCCCACACCA). The authors argue that this may change the affinity of Klu binding. Given that Klu is expressed in the R7, and that perturbations of Klu phenocopy sin, this is a plausible explanation. Nevertheless, no experimental validation is provided to show that Klu actually binds to this region and that the binding is affected by the insertion. I realise that ChIP-seq is difficult given the limited amount of material. Can another experiment be thought of? If not, can this be argued in the text – that this evidence is indirect, and no formal proof is provided that Klu binding in vivo is affected by this insertion. The authors provide surrogate evidence that Klu SELEX-seq is biased towards CGCCCACACCA sequences having sin; this is interesting but it is not convincing to me, because the SELEX logo's do not contain this flanking site, rather they suggest Klu binds to CGCCCACGCA. This is quite different from the site with sin (notice the trailing GCA). Given the very high conservation of a 17 bp stretch that contains a match to Klu, an alternative hypothesis is that actually two (or even more) transcription factors may bind to this 17bp stretch, perhaps Klu together with another factor, and that sin affects binding of the other, yet unknown factor, rather than of Klumpfuss. Klu perturbations would still affect Ss expression in that scenario, but not because of differential affinity with the site, but simply because Klu regulates Ss (in other words, Klu perturbations are not performed with a sin comparison, so they merely confirm that Klu regulates Ss). A possible experiment would be to perform the Klu perturbations with and without the sin insertion. Or to clone this region near an enhancer-reporter construct that activates a reporter, for example, in S2 cells (a strong enhancer that is active in S2 cells, many are known). Given the prediction of Klu-mediated repression, it can be investigated whether the reporter can be repressed by the sin-encompassing region, when transfecting with Klu cDNA, and whether this effect is changed between the wild type and the sin containing sequence. in vivo this would obviously be better, but would require more time. I am aware such experiments are challenging, so other experiments can be proposed by the authors, or if they are not possible given the limited time, the lack of formal evidence what this insertion does, and alternative hypotheses, should be discussed thoroughly (perhaps with a figure/cartoon).

We have followed the reviewer’s suggestion and performed Klu perturbations in flies with and without *sin.* Additionally, we proposed alternative hypotheses involving cooperative and/or competitive interactions between Klu and additional factors.

Increasing both binding affinity and Klu levels further decreases the ratio of Ss on/off expression: Since the proportion of Ss^ON^ R7s is reduced in flies with high Klu (high repressor) or in flies with the *sin* variant (high affinity), we predicted a further reduction in flies with both high Klu and the *sin* variant (high repressor, high affinity). We generated flies with ectopic expression of Klu in a *sin* genetic background and observed a significant additional reduction (Figure 4).

“Our data suggest that the ratio of Ss on/off gene expression is controlled by both the level of Klu protein and the binding affinity of the Klu site. […] We generated flies with increased levels of Klu in a sin genetic background and observed a significant additional reduction in the proportion of Ss^ON^ R7s (Figure 4).”

“Figure 4. Levels of Klu determine the ratio of Ss^ON^/Ss^OFF^ R7s […] In D, **** indicates p<0.0001. Error bars indicate standard deviation (SD).”

Lowering *klu* gene dosage suppresses the *sin* phenotype: An additional prediction is that lowering Klu levels in flies with *sin* should suppress the *sin* phenotype. We tested this prediction and found that reduced *klu* gene dosage suppressed the *sin* phenotype (Figure 4).

“To further test the relationship between Klu levels and binding site affinity, we reduced *klu* gene dosage in flies with *sin* and found that the *sin* phenotype was suppressed in *klu* mutant heterozygotes (Figure 4). We conclude that *sin* increases Klu binding affinity and that the binding affinity of the Klu site and levels of Klu protein determine the proportion of Ss^ON^ R7s.”

“Figure 4. Levels of Klu determine the ratio of Ss^ON^/Ss^OFF^ R7s. J. Decreasing *klu* gene dosage in *klu* null mutant heterozygotes suppressed the *sin* phenotype. ** indicates p<0.01. Error bars indicate standard deviation (SD).”

Alternative hypotheses:The following observations support the conclusion that *sin* increases the binding site affinity for Klu, causing a reduction in the proportion of Ss^ON^ R7s:

1) Selex-seq data show that *sin* increases the binding affinity of the endogenous low-affinity Klu site (also see response to reviewer point 5 below).

2) Changing the binding site to an optimized high-affinity Klu site mimics the *sin* phenotype.

3) Increasing Klu protein level mimics increasing Klu binding affinity in flies with *sin.*

4) Increasing both affinity and Klu levels causes an additional decrease in the proportion of Ss^ON^ R7s.

5) Decreasing *klu* gene dosage suppressed the effect of *sin.*

These results do not rule out other possible hypotheses involving cooperative and/or competitive interactions. We describe these additional hypotheses in the Discussion:

“The on/off nature of Ss expression suggests a cooperative mechanism whereby Klu acts with other factors to regulate ss. […] These activating transcription factors may compete for binding with the repressor Klu to determine the stochastic on/off expression state of ss.”

5) Given that the Klu binding site is not very informative (lots of C's), how significant is the prediction that this is a Klu binding site? On 100 random genes with similar size as Ss, how many Klu matches are found? Can this be reported? Does the score change when sin is included? That is, does the Klu PWM score increase with sin compared to control/null sites – likely not because sin is not present in the PWM, correct? Can this be discussed in the paper. This is important because the PWM is a measure for the binding affinity, which is argued to be increased with sin, so the PWM score should change (significantly?).

The reviewer’s comment concerns the nature of the *sin* change to the Klu binding site. We have addressed the reviewer’s concern by further analysis of the SELEX-seq data, specifically looking at the dependence of the contributions of individual bases to binding affinity.

We present these findings in (Figure 3) and describe them in the text and the figure legend:

“To evaluate the effect of *sin* on Klu binding, we analyzed available SELEX-seq binding data (Nitta et al. 2015), focusing on the core 10-mer. […] These data suggest that the Klu site in the endogenous locus is a low-affinity site and that *sin* increases its affinity.”

“Figure 3. *sin* increases the binding affinity for the transcription factor Klumpfuss […] Boxes highlight positions 8 and 10 in the 10-mer core sequences.”

6) It is written "As sin increases Klu binding affinity" – can this be changed into "As sin is predicted to increase Klu binding affinity"?

We have made the text change suggested by the review:

“As *sin* decreases Ss expression frequency and is predicted to increase Klu binding affinity, we hypothesized that Klu also represses stochastic ss expression in R7s.”

7) The role of the encompassing genomic region, how it is involved in the regulation of Ss, is not investigated – is this a new regulatory element of Ss? Are there any Janelia Flylight or VDRC tiles overlapping this region? If not, please discuss this. If there is time to clone this region into an enhancer-reporter and create a transgenic fly, that would be informative. If Flylight/VDRC lines are available, can they be tested with UAS-GFP to investigate whether this region is actually an Ss enhancer.

To address this question, we tested available Janelia Flylight lines that overlapped or neighbored this region and observed expression in the larval fly eye (Figure 1—figure supplement 1). This observation suggests that this region contains a regulatory DNA element that acts with the R7/R8 enhancer and two silencers to control stochastic Ss expression.

“The regions encompassing and neighboring the Klu binding site drive gene expression in the eye (Figure 1—figure supplement 1), suggesting that complex interactions between this regulatory DNA element, the R7/R8 enhancer, and the two silencers (Figure 1) ultimately control the ss on/off decision.”

“Figure 1—figure supplement 1. The regions encompassing and neighboring the Klu binding site have transcriptional activity in the eye […] The region encompassing the Klu binding site drive expression in the larval retina.”